# Primary Care Physicians’ Knowledge and Attitudes Regarding Palliative Care in Northeast Malaysia

**DOI:** 10.3390/healthcare11040550

**Published:** 2023-02-13

**Authors:** Norhazura Hamdan, Lili Husniati Yaacob, Nur Suhaila Idris, Mohd Shafik Abdul Majid

**Affiliations:** 1Department of Family Medicine, Universiti Sains Malaysia, Kubang Kerian 16150, Malaysia; 2Kemaman Health District, Ministry of Health Malaysia, Chukai Terengganu 24000, Malaysia

**Keywords:** primary healthcare, knowledge, attitude, palliative care

## Abstract

Palliative care in Malaysia has progressed steadily since its inception in 1991, and it has been integrated gradually into primary health care in the past decade. This study aims to assess the level of knowledge and the attitudes towards palliative care and its associated factors among primary care physicians. A cross-sectional study was conducted among primary care physicians using two validated questionnaires: the Palliative Care Knowledge Test (PCKT) and Frommelt’s Attitude Toward Care of the Dying (FATCOD). The data were analysed using descriptive and linear regression statistics. A total of 241 primary care physicians from 27 different health clinics participated in the study. The mean PCKT score was 8.68 (2.94), whereas the mean FATCOD score was 106.8 (9.14). The maximum score for each questionnaire was 20 and 150, respectively. There was a significant positive relationship between knowledge and attitudes toward palliative care, with a *p*-value of 0.003 (CI 0.22–1.04) and an r-value of 0.42. Palliative care knowledge among primary care physicians is still low despite their overall positive attitude towards the service. This finding suggests the urgent need for more education and training on palliative care for primary care physicians in Malaysia.

## 1. Introduction

Malaysia has a dire need for palliative care in oncology and other life-limiting illnesses. It is estimated that, by 2030, there will be an increase of 240% in palliative care needs compared to only 71% in 2014 [1,2]. Malaysia is also facing the prospect of having an aging population, whereby in 2030, the population of people aged 65 and older is estimated to be 15% [3]. This senior population, in which lives are longer and people have access to better healthcare, increases the demand for palliative care services. As a result of this demographic and epidemiological shift, the burden of non-communicable diseases (NCD) will also rise. In 2017, nine out of ten primary causes of death in Malaysia were reported to be NCD-related, and these patients should have received palliative care consultations [1].

Currently, patients with palliative needs are mainly managed in a hospital setting by a limited number of palliative specialists [4]. However, in the past decade, there has been a policy by the Ministry of Health to decentralise the management of palliative care from hospitals to health clinics or primary care clinics [4]. Primary care physicians must therefore increase their knowledge and abilities to provide palliative care as their expertise will be increasingly important given the needs of older populations. Primary palliative care includes fundamental skills, such as managing basic physical and psychological symptoms and discussing the goals of care; specialised palliative care includes the management of more complicated issues, such as refractory symptoms, existential distress, or family conflict. In an integrated palliative care model, primary providers deliver most of the palliative care and consult specialised physicians for complex or refractory problems, regardless of the stage of illness [5,6]. International studies have stressed how crucial it is for primary care doctors to be a vital part of the continuum of palliative care delivery in the community [2,7]. However, providing primary palliative care is not without its difficulties. These include the inadequacy of palliative care training, a negative attitude toward palliative care among primary care physicians, ambiguous roles and responsibilities, and a lack of structured communication and collaboration between primary care physicians and specialist physicians [8]. The aim of this study is to investigate the knowledge and attitudes of primary care physicians regarding palliative care.

## 2. Materials and Methods

### 2.1. Study Design

This study is a descriptive and correlational study.

### 2.2. Settings

This research was conducted in public primary health clinics in Kelantan. The state is in the northeast of Malaysia, with an area of 17,100 km^2^ that houses 1.91 million people. Health clinics are the backbone of public primary health care in Malaysia. Patient health care in clinics is provided by family medicine specialists and medical officers with the support of nurses and other paramedics. There are 107 health clinics in the state of Kelantan, however only 27 of them have in-house family medicine specialists. Each health clinic can have from five to thirty medical officers and one or two family medicine specialists, depending on the size of the clinic.

### 2.3. Participants

This study was a cross-sectional study conducted from 1 July 2020 to 1 November 2020. The sample size was calculated by comparing two proportions for the categorical variables and comparing two means for the numerical variables using version 3.1.2 of the Power Sample (PS) software. The largest sample size belonged to the variable “attendance to palliative seminars” [9], which yielded a sample size of 257 after considering 20% of non-responders.

This study utilized the convenience sampling method. All health clinics with in-house family medicine specialists (*n* = 27) in all 10 districts of Kelantan were chosen. All family medicine specialists and medical officers working on the day of data collection were invited to join in the study. In Malaysian healthcare settings, medical officers who work at health clinics are those who have at least completed their two years of compulsory service at government hospitals as medical interns. They are also those who have not undergone specialty training. For the purpose of this research, both the family medicine specialists and the medical officers working at the health clinics are identified as primary care physicians. Participants provided written informed consent to participate in this research prior to the data collection, and this study was approved by the Human Research Ethics Committee, Universiti Sains Malaysia (USM/JEPeM/19080462), and the Medical Research and Ethic Committee, Ministry of Health Malaysia (NMRR-1902596-50052(IIR)).

### 2.4. Measures

Knowledge was measured by adapting the Palliative Care Knowledge Test (PCKT) questionnaire [10]. The questionnaire has twenty items and is divided into five domains: philosophy (items 1 and 2), pain (items 3 to 8), dyspnoea (items 9 to 12), psychiatric problems (items 13 to 16), and gastrointestinal problems (items 17 to 20). Participants answer every question as either “true”, “false”, or “unsure”. One point is given for a correct answer and zero points for incorrect and “unsure” answers. The total score ranges from 0 to 20 and is converted into a mean and standard deviation (SD). A higher score indicates higher knowledge. The translated English version was used in this study and a few modifications were made to suit the local setting. Three items regarding specific drug use (pentazocine and buprenorphine hydrochloride) were substituted since these drugs are not widely used in Malaysia. Item number four, pentazocine, was changed to oxycodone as the latter is more commonly used for pain control in Malaysian settings. Item number six, which is “The effect of opioids should decrease when pentazocine or buprenorphine hydrochloride is used together after opioids are used”, was changed to “Even if breakthrough pain occurs when opioids are taken on a regular basis, the next dose should not be given earlier than scheduled.” Item number 12, “Anticholinergic drugs or scopolamine hydrobromide are effective for alleviating bronchial secretions of dying patients” was changed to “Evaluation of dyspnoea should be based on the subjective report of patients”. Preliminary testing of the questionnaire was done in a pilot study yielding a Cronbach alpha of 0.7.

Attitudes were measured using Frommelt Attitude Toward Care of the Dying-form B (FATCOD Form B) [11]. This form has been used and evaluated for validity and reliability, and it has demonstrated an interrater agreement of 1.0 and a Pearson’s Coefficient of 0.93 for test-retest reliability. FATCOD Form B consists of 30 Likert-type items scored on a five-point scale from 1 (Strongly Disagree), 2 (Disagree), 3 (Uncertain), 4 (Agree), to 5 (Strongly Agree). The instrument is made up of an equal number of positively and negatively worded items. Items 1, 2, 4, 16, 18, 20, 21, 22, 23, 24, 25, 27, and 30 are positively-worded statements. All others were negatively worded. The total score, ranging from 30 to 150, was converted into a mean and standard deviation with higher scores indicating more positive attitudes.

## 3. Results

A total of 241 primary care physicians participated in this study. The participants’ mean (SD) age was 33.41 (5.48) years old, with a range from 27 to 55 years old. The majority of the respondents were female (73.4%). Table 1 shows that medical officers made up 224 of the total 241 responders, while just 17 of responders were practicing as family specialists, accounting for 92.9 percent and 7.1 percent, respectively. Their mean (SD) service time was 7.65 (5.48) years.

Only 12.6% of the respondents had attended a palliative seminar before, and 30.8% had worked in a hospice or palliative care unit. A total of 47.7% of those who took part indicated that they had never encountered a palliative situation previously. The mean (SD) number of palliative cases ever seen was 14.83 (20.16), with 69% of participants not having encountered any palliative cases in the preceding year.

Based on the results of the PCKT test, the mean knowledge score (SD) was 8.68 (2.94). The domain of pain had the best score, a mean (SD) of 3.07 (1.22), with philosophy having the lowest score, with a mean (SD) of 1.27 (0.68). As illustrated in Table 2, more than 90% of the participants correctly answered the question about the role of pain control in good sleep. However, only 17% of participants knew the association between opioids and addiction.

Based on FATCOD, the mean (SD) score for attitudes was 106.8 (9.14). When the reports were broken down by item, the following statements had the highest mean score for the entire sample, as shown in Table 3: statement 1 (Giving care to the dying person is a worthwhile experience) had a mean of 4.46 and a SD of 0.71, and statement 18 (Families should be concerned about helping their dying member make the best of his/her remaining life) had a mean of 4.46 and a SD of 0.66). The statements with the lowest mean scores were statement 10(There are times when the dying person welcomes death), which had a mean of 2.10 and a SD of 0.63, and statement 12 (The family should be involved in the physical care of the dying person) which had a mean of 1.43 and a SD of 0.74.

There was a statistically significant relationship between respondent knowledge and attitudes toward palliative care (*p* = 0.003 and an r value of 0.42). A one-point rise in the mean knowledge score resulted in a 0.63-unit increase in positive attitudes towards palliative care (95% CI 0.22–1.04). Other factors such as age, gender, years of service, number of patients seen, seminar attendance, and exposure to a palliative care unit, on the other hand, had no association with attitudes on palliative care.

## 4. Discussion

We determined that the PCKT (knowledge) score was quite low, being 8.68 out of a maximum of 20. The pain component was shown to be higher than any of the other components. This could be related to participants’ knowledge on general pain management not necessarily related to palliative care. Most responders, on the other hand, did not have a high score on any other aspect of the palliative care components. The results were similar to the original study conducted in both physicians and nurses, with a mean (SD) of 8.2 (4.3), and with the highest score being attained in the same pain component [10]. In a study conducted in Spain, the same pain domain was shown to have the highest score, while the entire PCKT-SV (Spanish version) mean (SD) score was 10.7 (3.2) [12].

Although the pain domain had the highest score, the finding of the study regarding knowledge on opioid use in palliative care is worrying. The majority of our participants did not have a strong understanding regarding the use of opioids in palliative care, as shown by the fact that less than half of the respondents correctly answered opioid-related questions. Of note is the fact that about 80% of the participants thought that long-term opioid use can cause addiction in palliative patients. The lack of access to opioids in the health clinics in Malaysia could have contributed to the poor understanding of opioid use. This limited access may have contributed to the respondents’ lack of awareness and understanding of pharmacotherapy and drug side effects and to their unwillingness to prescribe opioids because of their lack of expertise [13]. This limited access may also have also contributed to the inadequate training on opioid use, which is shown by a study conducted among physicians in ten Asian nations, including Malaysia, whereby 30.5 percent of physicians believed their medical school teaching on opioid usage was inadequate [14].

Our research revealed that the philosophy domain had the lowest score, with a mean of 1.27. This is similar to the Spanish study, which stated that continuous training was a good predictor of the philosophy domain score [12]. The philosophy of palliative care is a focus on the quality of life rather than on a cure. It affirms life and views death as a natural process. The focus of treatment is on the alleviation of pain and other unpleasant symptoms with an emphasis on the psychological and spiritual aspects of care [15]. Palliative care also provides patients with a support system so that they can live as fully as possible until death and so that families can be assisted during the patient’s illness and during bereavement [15]. It is crucial for primary care professionals to understand this core principle of palliative care in order to provide better care for patients with terminal illness. Understanding the idea of palliative care also gives primary care clinicians the confidence to communicate with patients and their families more effectively.

We determined that the mean (SD) score of FATCOD, which measures attitudes, was 106.8 (9.14). The score in our study was lower compared to the score of 118 in the original intervention study evaluating nurses’ attitudes towards terminally ill patients [11]. Furthermore, a study of first-year nursing students in Sweden revealed that the FATCOD mean (SD) score was 119.5 (10.6), whereas a prior study of second- and third-year nursing students in the same nation found that the FATCOD mean (SD) score was 127.8 (8.1) [16]. Another study, which was done in Saudi Arabia among multinational nurses using the same tool, also noted a higher score of 111.66 (13.97) [17].

We found that the item scoring the lowest in the attitude scoring regarded the involvement of family members in the physical care of a dying person. The involvement of family members is crucial in the care of patients with palliative needs. As cancer progresses, patients will invariably depend on their caregivers to look after their needs, in terms of the activities of daily living such as eating and bathing. Caregivers will also need to look after patients’ medical needs such as giving medication and dressing wounds. The involvement of family members is not limited to caring for the medical and non-medical needs of patients. A considerable proportion of seriously ill patients lose the capacity to make some or all of their medical decisions at the end of their lives [18]. It is morally and legally acceptable in these situations to delegate the decision-making authority to close family members or friends. Therefore, it is crucial for health professionals to acknowledge the latter individuals’ roles and involve them as part of the team in the care of the dying.

Our study has shown that there is a positive correlation between attitudes and knowledge regarding palliative care. The most significant factor identified in our study that was related to higher attitude scores was knowledge on palliative care. Knowledge has been shown in many other studies to be closely related to attitudes [17,19]. In a study on primary care physicians in Kuwait, the majority of physicians had poor knowledge of palliative care and, as a result, had unfavourable attitudes towards palliative care [19]. The area regarding the improvement of education and the training of primary healthcare professionals on palliative care has been extensively researched and discussed [20]. Multiple studies have shown the need for primary health care professionals to improve their knowledge and skills in managing patients with palliative needs [21]. There are multiple strategies that can be employed to improve the education and training on palliative care among primary health care professionals. Currently, the teaching on palliative care to medical students in Malaysia is inadequate [22,23]. Therefore, it is first and foremost necessary to improve the undergraduate teaching on palliative care of medical students [24]. Similar to the exposure to palliative care during undergraduate teaching, the teaching and training on palliative care in postgraduate specialty programs of family medicine at university also need to be improved [25]. Among the areas which require particular attention in the postgraduate training is the area of interprofessional or community coordination between all the teams responsible for palliative care in the community [20]. Furthermore, a systemic review on a palliative care training programme in Latin America suggested that practice-based methods and exposure to clinical settings should be integrated into ongoing courses to facilitate learning. In addition, the creation of a regional platform is needed to share experiences of successful training programs and foster the development of palliative care education [26].

## 5. Conclusions

This study has shown that the knowledge on palliative care of our primary health care doctors were still inadequate. More concerted efforts have to be applied to ensure training on palliative care is integrated in education programs. Improvements in palliative care education will not only provide benefits for dying patients and their families but will also extend to the care of many other primary care patients, including geriatric patients and those with chronic illnesses, who make up a large proportion of the adult primary care population.

## Figures and Tables

**Table 1 healthcare-11-00550-t001:** Socio-demographics, health service characteristics, and exposure to palliative service.

	Mean	(SD)	N	(%)
**Socio-demographics**				
**Age**	33.41	5.48		
21–30			78	32.4
31–40			139	57.7
41–50			22	9.1
>50			2	0.8
**Gender**				
Male			64	26.6
Female			177	73.4
**Religion**				
Muslim			234	97.1
Buddhist			7	2.9
**Health service**				
Profession				
Family Physician			17	7.1
Medical Officer			224	92.9
**Duration of service, years**	7.65	5.48		
<1			13	5.5
1–4			61	26.0
5–10			118	50.2
11–15			20	8.5
16–20			11	4.7
>20			12	5.1
**Attendance in palliative seminars**				
Yes			30	12.6
No			209	87.4
**Exposure to palliative unit/hospice**				
Yes			74	30.8
No			166	69.2
**Number of palliative cases seen since service**	14.83	20.16		
None			115	47.7
1–10			87	36.1
11–50			35	14.5
51–100			4	1.7
>100			0	0
**Number of palliative cases seen past one year**	4.05	4.60		
None			166	68.9
1–10			70	29.0
10–50			5	2.1
>50			0	0

**Table 2 healthcare-11-00550-t002:** Results of Palliative Care Knowledge Test (PCKT).

Items	Correct Answer	
	N	(%)
**Philosophy**		
Palliative care should only be provided for patients who have no curative treatments available.	102	42.5
Palliative care should not be provided along with anti-cancer treatments.	202	84.2
**Pain**		
**One of the goals of pain management is to get a good night’s sleep**	**220**	**91.3**
When cancer pain is mild, oxycodone should be used more often than an opioid.	75	31.4
When opioids are taken on a regular basis, nonsteroidal anti-inflammatory drugs should not be used.	102	42.5
Even if breakthrough pain occurs when opioids are taken on a regular basis, the next dose should not be given earlier than scheduled.	121	50.4
Long-term use of opioids can often induce addiction.	41	17.2
**Use of opioids does not influence survival time.**	**171**	**72.5**
**Dyspnoea**		
**Morphine should be used to relieve dyspnoea in cancer patients.**	**66**	**27.6**
When opioids are taken on a regular basis, respiratory depression will be common.	71	30.2
Oxygen saturation levels are correlated with dyspnea.	53	22.3
**Evaluation of dyspnoea should be based on subjective report of patients.**	**129**	**53.8**
**Psychiatric problems**		
**During the last days of life, drowsiness associated with electrolyte imbalance should decrease patient discomfort.**	**65**	**27.3**
**Benzodiazepines should be effective for controlling delirium.**	**124**	**51.7**
**Some dying patients will require continuous sedation to alleviate suffering.**	**164**	**68.6**
Morphine is often a cause of delirium in terminally ill cancer patients.	60	25.0
**Gastrointestinal problems**		
At terminal stages of cancer, higher calorie intake is needed compared to initial stages.	76	31.8
There is no route except central venous for patients unable to maintain a peripheral intravenous route.	82	34.2
**Steroids should improve appetite among patients with advanced cancer.**	**53**	**22.1**
**Intravenous infusion will not be effective for alleviating dry mouth in dying patients.**	**99**	**41.3**

Correct statement in bold.

**Table 3 healthcare-11-00550-t003:** Results of Frommelt Attitude Toward Care of the Dying.

Items	Mean	(SD)
Giving care to the dying person is a worthwhile experience.	4.46	0.71
Death is not the worst thing that can happen to a person.	3.19	1.34
I would be uncomfortable talking about impending death with the dying person.	2.35	1.06
Caring for the patient’s family should continue throughout the period of grief and bereavement.	4.40	0.69
I would not want to care for a dying person.	4.04	0.89
The nonfamily caregivers should not be the one to talk about death with the dying person.	3.15	1.20
The length of time required giving care to a dying person would frustrate me	3.60	0.98
I would be upset when the dying person I was caring for gave up hope of getting better.	2.67	1.13
It is difficult to form a close relationship with the dying person.	3.27	1.05
There are times when the dying person welcomes death.	2.10	0.63
When a patient asks, “Am I dying?” I think it is best to change the subject to something cheerful.	3.05	1.13
The family should be involved in the physical care of the dying person.	1.43	0.74
I would hope the person I am caring for dies when I am not present.	3.52	1.03
I am afraid to become friends with a dying person.	3.79	0.98
I would feel like running away when the person died.	3.80	0.98
Families need emotional support to accept the behaviour changes of the dying person.	4.44	0.70
As a patient nears death, the nonfamily caregiver should withdraw from his/her involvement with the patient.	3.55	1.07
Families should be concerned about helping their dying member make the best of his/her remaining life.	4.46	0.66
The dying person should not be allowed to make decisions about his/her physical care	3.93	0.99
Families should maintain as normal an environment as possible for their dying member.	4.08	0.78
It is beneficial for the dying person to verbalize his/her feelings.	4.39	0.64
Care should extend to the family of the dying person.	4.28	0.70
Caregivers should permit dying persons to have flexible visiting schedules.	4.06	0.82
The dying person and his/her family should be the in-charge decision-makers.	4.04	0.80
Addiction to pain relieving medication should not be a concern when dealing with a dying person.	3.38	1.18
I would be uncomfortable if I entered the room of a terminally ill person and found him/her crying.	2.67	1.08
Dying persons should be given honest answers about their condition.	4.05	0.72
Educating families about death and dying is not a nonfamily caregiver responsibility	3.69	0.95
Family members who stay close to a dying person often interfere with the professional’s job with the patient.	2.92	0.92
It is possible for nonfamily caregivers to help patients prepare for death.	3.85	0.73

## Data Availability

Not applicable.

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
