# Peer review of "Primary Care Physicians’ Knowledge and Attitudes Regarding Palliative Care in Northeast Malaysia"

_healthcare, 2023, doi:10.3390/healthcare11040550_

Round 1
Reviewer 1 Report
The authors administered knowledge and philosophy based questionnaires to primary health care providers in Malaysia assessing palliative care. They report generally low scores, underscoring the need for improved palliative care education for primary health care providers. The manuscript is well written and succinct. It makes it point clearly
There are a few easily fixable misspellings and grammatical errors (line 35)
Author Response
I have made the necessary amendment as required

Reviewer 2 Report
Thank you for the opportunity to review this manuscript.
1) OVERALL: Identification of the Sample: Need to standardize naming of the group of participants for this study and be consistent (line 2 Primary health care doctors; abstract line15, primary healthcare workers; line 21 primary health care professionals; line 37 primary care physicians; line 46 primary care doctors; line 52 primary healthcare doctors; line 97 family physicians; line 105 states 242 doctors).
Suggest : Primary health care physicians' knowledge and attitudes regarding palliative care in Northeast Malaysia
OR change title to "The Knowledge and Attitudes of Family Medicine Physicians towards Palliative Care in Northeast Malaysia"
2) Edits Suggested for the Introduction:
Line 26: .. there will be an increase of 240%..
Line 28 : people aged 65 o older
Line 32: In 2017, nine out of ten primary causes of death in Malaysia were reported to be NCD-related, who should receive a palliative care consultation.
Line 35: "..a limited number of palliative specialists..
Line 36: Upper case Ministry of Health
Line 38: Primary care physicians must therefore increase their knowledge and abilities to provide palliative care as their expertise will be increasingly important given the needs of older populations.
EDITS TO MATERIALS and METHODS
Line 54: Authors need to identify the study design before Setting. The study design is a descriptive and correlational study.
Following design, setting, participants, the authors should then discuss the Measures.
Line 80: Attitudes were measured using ...
Line 87: The total score, ranging from 30 to 150, were converted into a mean and standard deviation with higher scores indicating more positive attitudes.
Line 92: PARTICIPANTS--not clear how the sample size was calculated. The usual approach is to use Cohen's power analysis. Line 92 to 94: please clarify the approach reported . Line 94- largest sample size was from variable "Attendance at Seminars." Not clear how this relates to determination of appropriate sample size for this study.
Line 97: Clarity is needed.. Suggestion "All health clinics (n = 27) with family physicians. If you put (N=27) after physicians it is confusing as it indicates that 27 physicians participated.
Line 98: what is the difference between a family physicians in these clinics from the medical officers (especially since the majority of the sample are medical officers). Need to explain if the medical officers are family physicians. Do the medical officers provide direct patient care?
EDITS TO RESULTS:
Line 105: A total of 242 physicians participated in this study.
**Add information about Gender of participants.
Line 109: "Length of their medical experience ranged from six months to thirty years" --THIS IS NOT DISCUSSED IN TABLE 1:
In Table 1: it is not clear if the duration of service in years represents what is identified in Line 109 as discussed above.
Table 1: needs editing "Socio-demographics
Bold age, gender, religion.
Under Health Services: Delineate Health Services Profession and bold also duration of service (but clarify what this means)
Lines 115: Need to name the instruments more specifically when discussing results.
Line 116: Based on the results of the PCKT test, the mean knowledge score...
Line 122: Based on the FATCOD, the means (SD) score for attitudes was>>>
Table 2: name should be more clear (Results of Palliative Care Knowledge Test (PCKT)
Suggest that in Table 2: for reader information suggest identifying the correct answer.
In Table 2: It would be helpful to group the items according to the domains (Philosophy, Pain, Dyspnea, GI, Psych) identified under "measures."
Table 3: rename the title to (Results of Frommelt Attitude Toward Care of the Dying)
The total score on this measure ranges from 30 to 150. Suggest reporting the MEAN SCORE for the instrument and what the score represents. Then identifying the mean score for each item.
Clarity needs to be given to what the mean score and SD represents for each item. What does the mean score for each item represent given the possible range of scores.
SUGGESTED EDITS TO DISCUSSION--overall adequate
Overall: Suggest discussing the findings on the PCKT according to the domains identified (Philosophy, Pain, Dyspnea, Psych, GI)
Line 138 : We determined that the PCKT (knowledge) score was quite low... (identify the Instrument)
Line 157-160: Check font and font size
Line 173: "We determined that the mean (DS) score on the FATCOD, measuring attitudes, were 106.8 (9. 14).
Line 193: Need to more fully emphasize the results of the correlation between PCKT and FATCOD
Line 203: Medical students in most countries are graduate students. Not clear why the authors discuss "Undergraduate medical students."
EDITS TO CONCLUSION
Suggest more in-depth discussion of strategies to increase primary care physicians' Knowledge and Attitudes Toward Palliative Care.
Please identify specific recommendations to increase physicians' knowledge of palliative care from graduate medical education to post graduate education.
I hope these suggestions will strengthen the manuscript.
Author Response
I have made the necessary amendments as attached in the document

Reviewer 3 Report
In their study, Hamdan et al. show that primary healthcare workers have only poor knowledge of palliative care, and hence, they see an urgent need to improve education in this crucial field. The study is well-conducted and easy to read. I also appreciate it very much that they drew comparisons to other countries such as Sweden or Saudi Arabia, which underlines that the analyzed issue is not only limited to Malaysia but of global importance. By drawing concrete comparisons, they moreover, were able to objectively and quantitatively address the lack of knowledge in Malaysian palliative care. I only have two minor remarks:
1. What do you mean by caspecialistsist (line 35)? I do not know this word and I assume that it is a type error. Please replace it by the correct term.
2. Why did you enroll so many medical officers in your study? Is this a particularity of Malaysia? Please describe the role of medical officers in your paper and the reason why they are so over-represented in your study. This will be definitely of help to international readers.
Author Response

(The authors gave the same response as above.)

Round 2
Reviewer 2 Report
Dear Authors,
Thank you for the revisions made to the manuscript. In the file attached below, I highlighted in yellow any areas that need some edits. For example, in the US we say "Primary care physicians" we do not add the word "health" such as Primary healthcare physicians. Perhaps this is different in your country.
Other areas are highlighted in yellow because the word needs to be plural such as adding an "s" to specialists.
Add the word "The" to 'The majority.."
Question regarding medical officers--they have graduated from medical school but what further education do they have after graduation to prepare them to practice as generalists? Can you provide more information on how their preparation is different from the family medicine specialists?
Please just make the small suggested edits. As I could not type on the document, please review the yellow highlights and have a colleague read the sentence to identify the small suggested edits.
Best Wishes!

Author Response
I have made the necessary edits
